# Maximizing Subset Accuracy with Recurrent Neural Networks in Multi-label Classification

**Jinseok Nam[1], Eneldo Loza Mencía[1], Hyunwoo J. Kim[2], and Johannes Fürnkranz[1]**

[1]Knowledge Engineering Group, TU Darmstadt
[2]Department of Computer Sciences, University of Wisconsin-Madison

## Abstract

Multi-label classification is the task of predicting a set of labels for a given input instance. Classifier chains are a state-of-the-art method for tackling such problems, which essentially converts this problem into a sequential prediction problem, where the labels are first ordered in an arbitrary fashion, and the task is to predict a sequence of binary values for these labels. In this paper, we replace classifier chains with recurrent neural networks, a sequence-to-sequence prediction algorithm which has recently been successfully applied to sequential prediction tasks in many domains. The key advantage of this approach is that it allows to focus on the prediction of the positive labels only, a much smaller set than the full set of possible labels. Moreover, parameter sharing across all classifiers allows to better exploit information of previous decisions. As both, classifier chains and recurrent neural networks depend on a fixed ordering of the labels, which is typically not part of a multi-label problem specification, we also compare different ways of ordering the label set, and give some recommendations on suitable ordering strategies.

## 1 Introduction

There is a growing need for developing scalable *multi-label classification* (MLC) systems, which, e.g., allow to assign multiple topic terms to a document or to identify objects in an image. While the simple *binary relevance* (BR) method approaches this problem by treating multiple targets independently, current research in MLC has focused on designing algorithms that exploit the underlying label structures. More formally, MLC is the task of learning a function $f$ that maps inputs to subsets of a label set $\mathcal{L} = \{1, 2, \cdots, L\}$. Consider a set of $N$ samples $\mathcal{D} = \{(\boldsymbol{x}_n, \boldsymbol{y}_n)\}_{n=1}^{N}$, each of which consists of an input $\boldsymbol{x} \in \mathcal{X}$ and its target $\boldsymbol{y} \in \mathcal{Y}$, and the $(\boldsymbol{x}_n, \boldsymbol{y}_n)$ are assumed to be *i.i.d* following an unknown distribution $P(\boldsymbol{X}, \boldsymbol{Y})$ over a sample space $\mathcal{X} \times \mathcal{Y}$. We let $T_n = |\boldsymbol{y}_n|$ denote the size of the label set associated to $\boldsymbol{x}_n$ and $C = \frac{1}{N} \sum_{n=1}^{N} T_n$ the cardinality of $\mathcal{D}$, which is usually much smaller than $L$. Often, it is convenient to view $\boldsymbol{y}$ not as a subset of $\mathcal{L}$ but as a binary vector of size $L$, i.e., $\boldsymbol{y} \in \{0, 1\}^L$. Given a function $f$ parameterized by $\theta$ that returns predicted outputs $\hat{\boldsymbol{y}}$ of inputs $\boldsymbol{x}$, i.e., $\hat{\boldsymbol{y}} \leftarrow f(\boldsymbol{x}; \theta)$, and a loss function $\ell : (\boldsymbol{y}, \hat{\boldsymbol{y}}) \rightarrow \mathbb{R}$ which measures the discrepancy between $\boldsymbol{y}$ and $\hat{\boldsymbol{y}}$, the goal is to find an optimal parametrization $f^*$ that minimizes the expected loss on an unknown sample drawn from $P(\boldsymbol{X}, \boldsymbol{Y})$ such that $f^* = \arg\min_f \mathbb{E}_{\boldsymbol{X}} \left[ \mathbb{E}_{\boldsymbol{Y}|\boldsymbol{X}} \left[ \ell(\boldsymbol{Y}, f(\boldsymbol{X}; \theta)) \right] \right]$. While the expected risk minimization over $P(\boldsymbol{X}, \boldsymbol{Y})$ is intractable, for a given observation $\boldsymbol{x}$ it can be simplified to $f^*(\boldsymbol{x}) = \arg\min_f \mathbb{E}_{\boldsymbol{Y}|\boldsymbol{X}} \left[ \ell(\boldsymbol{Y}, f(\boldsymbol{x}; \theta)) \right]$. A natural choice for the loss function is *subset 0/1 loss* defined as $\ell_{0/1}(\boldsymbol{y}, f(\boldsymbol{x}; \theta)) = \mathbb{I}[\boldsymbol{y} \neq \hat{\boldsymbol{y}}]$ which is a generalization of the 0/1 loss in binary classification to multi-label problems. It can be interpreted as an objective to find the mode of the joint probability of label sets $\boldsymbol{y}$ given instances $\boldsymbol{x}$: $\mathbb{E}_{\boldsymbol{Y}|\boldsymbol{X}} \left[ \ell_{0/1}(\boldsymbol{Y}, \hat{\boldsymbol{y}}) \right] = 1 - P(\boldsymbol{Y} = \boldsymbol{y}|\boldsymbol{X} = \boldsymbol{x})$. Conversely, $1 - \ell_{0/1}(\boldsymbol{y}, f(\boldsymbol{x}; \theta))$ is often referred to as *subset accuracy* in the literature.

## 2  Subset Accuracy Maximization in Multi-label Classification

For maximizing subset accuracy, there are two principled ways for reducing a MLC problem to multiple subproblems. The simplest method, *label powerset* (LP), defines a set of all possible label combinations $\mathcal{S}_L = \{\{1\}, \{2\}, \cdots, \{1, 2, \cdots, L\}\}$, from which a new class label is assigned to each label subset consisting of *positive* labels in $\mathcal{D}$. LP, then, addresses MLC as a multi-class classification problem with $\min(N, 2^L)$ possible labels such that

$$P(y_1, y_2, \cdots, y_L | \boldsymbol{x}) \xrightarrow{LP} P(y_{\text{LP}} = k | \boldsymbol{x}) \tag{1}$$

where $k \in \{1, 2, \cdots, \min(N, 2^L)\}$. While LP is appealing because most methods well studied in multi-class classification can be used, training LP models becomes intractable for large-scale problems with an increasing number of labels in $\mathcal{S}_L$. Even if the number of labels $L$ is small enough, the problem is still prone to suffer from data scarcity because each label subset in LP will in general only have a few training instances. An effective solution to these problems is to build an ensemble of LP models learning from randomly constructed small label subset spaces [29].

An alternative approach is to learn the joint probability of labels, which is prohibitively expensive due to $2^L$ label configurations. To address such a problem, Dembczyński et al. [3] have proposed *probabilistic classifier chain* (PCC) which decomposes the joint probability into $L$ conditional probabilities:

$$P(y_1, y_2, \cdots, y_L | \boldsymbol{x}) = \prod_{i=1}^{L} P(y_i | \boldsymbol{y}_{<i}, \boldsymbol{x}) \tag{2}$$

where $\boldsymbol{y}_{<i} = \{y_1, \cdots, y_{i-1}\}$ denotes a set of labels that precede a label $y_i$ in computing conditional probabilities, and $\boldsymbol{y}_{<i} = \emptyset$ if $i = 1$. For training PCCs, $L$ functions need to be learned independently to construct a probability tree with $2^L$ leaf nodes. In other words, PCCs construct a perfect binary tree of height $L$ in which every node except the root node corresponds to a binary classifier. Therefore, obtaining the exact solution of such a probabilistic tree requires to find an optimal path from the root to a leaf node. A naïve approach for doing so requires $2^L$ path evaluations in the inference step, and is therefore also intractable. However, several approaches have been proposed to reduce the computational complexity [4, 13, 24, 19].

Apart from the computational issue, PCC has also a few fundamental problems. One of them is a cascadation of errors as the length of a chain gets longer [25]. During training, the classifiers $f_i$ in the chain are trained to reduce the errors $\mathcal{E}(y_i, \hat{y}_i)$ by enriching the input vectors $\boldsymbol{x}$ with the corresponding previous true targets $\boldsymbol{y}_{<i}$ as additional features. In contrast, at test time, $f_i$ generates samples $\hat{y}_i$ or estimates $P(\hat{y}_i | \boldsymbol{x}, \hat{\boldsymbol{y}}_{<i})$ where $\hat{\boldsymbol{y}}_{<i}$ are obtained from the preceding classifiers $f_1, \cdots, f_{i-1}$.

Another key limitation of PCCs is that the classifiers $f_i$ are trained independently according to a fixed label order, so that each classifier is only able to make predictions with respect to a single label in a chain of labels. Regardless of the order of labels, the product of conditional probabilities in Eq. (2) represents the joint probability of labels by the chain rule, but in practice the label order in a chain has an impact on estimating the conditional probabilities. This issue was addressed in the past by ensemble averaging [23, 3], ensemble pruning [17] or by a previous analysis of the label dependencies, e.g., by Bayes nets [27], and selecting the ordering accordingly. Similar methods learning a global order over the labels have been proposed by [13], who use kernel target alignment to order the chain according to the difficulty of the single-label problems, and by [18], who formulate the problem of finding the globally optimal label order as a dynamic programming problem. Aside from PCC, there has been another family of probabilistic approaches to maximizing subset accuracy [9, 16].

## 3  Learning to Predict Subsets as Sequence Prediction

In the previous section, we have discussed LP and PCC as a means of subset accuracy maximization. Note that $y_{\text{LP}}$ in Eq. (1) denotes a set of positive labels. Instead of solving Eq. (1) using a multi-class classifier, one can consider predicting all labels individually in $y_{\text{LP}}$, and interpret this approach as a way of maximizing the joint probability of a label subset given the number of labels $T$ in the subset. Similar to PCC, the joint probability can be computed as product of conditional probabilities, but unlike PCC, only $T \ll L$ terms are needed. Therefore, maximizing the joint probability of *positive* labels can be viewed as subset accuracy maximization such as LP in a sequential manner as the

way PCC works. To be more precise, $\boldsymbol{y}$ can be represented as a set of 1-of-$L$ vectors such that $\boldsymbol{y} = \{\mathbf{y}_{p_i}\}_{i=1}^T$ and $\mathbf{y}_{p_i} \in \mathbb{R}^L$ where $T$ is the number of positive labels associated with an instance $\boldsymbol{x}$. The joint probability of *positive* labels can be written as

$$P(\mathbf{y}_{p_1}, \mathbf{y}_{p_2}, \cdots, \mathbf{y}_{p_T}|\boldsymbol{x}) = \prod_{i=1}^T P(\mathbf{y}_{p_i}|\boldsymbol{y}_{<p_i}, \boldsymbol{x}). \tag{3}$$

Note that Eq. (3) has the same form with Eq. (2) except for the number of output variables. While Eq. (2) is meant to maximize the joint probability over the entire $2^L$ configurations, Eq. (3) represents the probability of sets of positive labels and ignores negative labels. The subscript $p$ is omitted unless it is needed for clarity. A key advantage of Eq. (3) over the traditional multi-label formulation is that the number of conditional probabilities to be estimated is dramatically reduced from $L$ to $T$, improving scalability. Also note that each estimate itself again depends on the previous estimates. Reducing the length of the chain might be helpful in reducing the cascading errors, which is particularly relevant for labels at the end of the chain. Having said that, computations over the $L^T$ search space of Eq. (3) remain infeasible even though our search space is much smaller than the search space of PCC in Eq. (2), $2^L$, since the label cardinality $C$ is usually very small, i.e., $C \ll L$.

As each instance has a different value for $T$, we need MLC methods capable of dealing with a different number of output targets across instances. In fact, the idea of predicting positive labels only has been explored for MLC. *Recurrent neural networks* (RNNs) have been successful in solving complex output space problems. In particular, Wang et al. [31] have demonstrated that RNNs provide a competitive solution on MLC image datasets. Doppa et al. [6] propose *multi-label search* where a heuristic function and cost function are learned to iteratively search for elements to be chosen as positive labels on a binary vector of size $L$. In this work, we make use of RNNs to compute $\prod_{i=1}^T P(\mathbf{y}_{p_i}|\boldsymbol{y}_{<p_i}, \boldsymbol{x})$ for which the order of labels in a label subset $\mathbf{y}_{p_1}, \mathbf{y}_{p_2}, \cdots, \mathbf{y}_{p_T}$ need to be determined a priori, as in PCC. In the following, we explain possible ways of choosing label permutations, and then present three RNN architectures for MLC.

### 3.1 Determining Label Permutations

We hypothesize that some label permutations make it easier to estimate Eqs. (2) and (3) than others. However, as no ground truth such as relevance scores of each positive label to a training instance is given, we need to make the way to prepare fixed label permutations during training.

The most straightforward approach is to order positive labels by frequency simply either in a descending (from frequent to rare labels) or an ascending (from rare to frequent ones) order. Although this type of label permutation may break down label correlations in a chain, Wang et al. [31] have shown that the descending label ordering allows to achieve a decent performance on multi-label image datasets. As an alternative, if additional information such as label hierarchies is available about the labels, we can also take advantage of such information to determine label permutations. For example, assuming that labels are organized in a *directed acyclic graph* (DAG) where labels are partially ordered, we can obtain a total order of labels by topological sorting with *depth-first search* (DFS), and given that order, target labels in the training set can be sorted in a way that labels that have same ancestors in the graph are placed next to each other. In fact, this approach also preserves partial label orders in terms of the co-occurrence frequency of a child and its parent label in the graph.

### 3.2 Label Sequence Prediction from Given Label Permutations

A *recurrent neural network* (RNN) is a *neural network* (NN) that is able to capture temporal information. RNNs have shown their superior performance on a wide range of applications where target outputs form a sequence. In our context, we can expect that MLC will also benefit from the reformulation of PCCs because the estimation of the joint probability of only positive labels as in Eq. (3) significantly reduces the length of the chains, thereby reducing the effect of error propagation.

A RNN architecture that learns a sequence of $L$ *binary* targets can be seen as a NN counterpart of PCC because its objective is to maximize Eq. (2), just like in PCC. We will refer to this architecture as RNN$^b$ (Fig. 1b). One can also come up with a RNN architecture maximizing Eq. (3) to take advantage of the smaller label subset size $T$ than $L$, which shall be referred to as RNN$^m$ (Fig. 1c). For learning RNNs, we use *gated recurrent units* (GRUs) which allow to effectively avoid the vanishing gradient problem [2]. Let $\bar{\mathbf{x}}$ be the fixed input representation computed from an instance $\boldsymbol{x}$. We shall

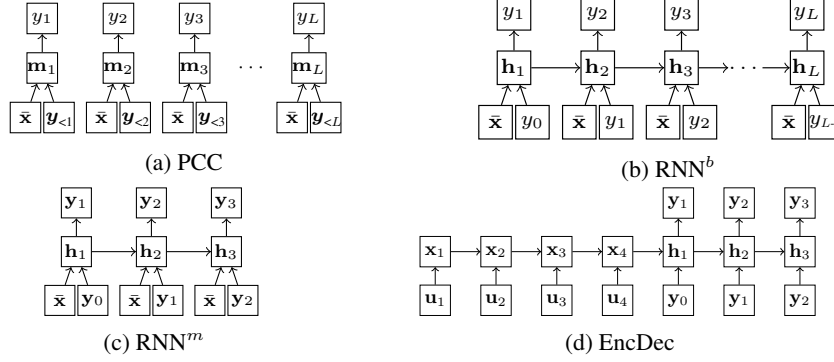

Figure 1: Illustration of PCC and RNN architectures for MLC. For the purpose of illustration, we assume $T = 3$ and $\boldsymbol{x}$ consists of 4 elements.

explain how to determine $\bar{\mathbf{x}}$ in Sec. 4.2. Given an initial state $\mathbf{h}_0 = f_{\text{init}}(\bar{\mathbf{x}})$, at each step $i$, both $\text{RNN}^b$ and $\text{RNN}^m$ compute a hidden state $\mathbf{h}_i$ by taking $\bar{\mathbf{x}}$ and a target (or predicted) label from the previous step as inputs: $\mathbf{h}_i = \text{GRU}\left(\mathbf{h}_{i-1}, \mathbf{V}_{y_{i-1}}, \bar{\mathbf{x}}\right)$ for $\text{RNN}^b$ and $\mathbf{h}_i = \text{GRU}\left(\mathbf{h}_{i-1}, \mathbf{V}\mathbf{y}_{p_{i-1}}, \bar{\mathbf{x}}\right)$ for $\text{RNN}^m$ where $\mathbf{V}$ is the matrix of $d$-dimensional label embeddings. In turn, $\text{RNN}^b$ computes the conditional probabilities $P_\theta\left(y_i | \boldsymbol{y}_{<i}, \boldsymbol{x}\right)$ in Eq. (2) by $f\left(\mathbf{h}_i, \mathbf{V}_{y_{i-1}}, \bar{\mathbf{x}}\right)$ consisting of linear projection, followed by the softmax function. Likewise, we consider $f\left(\mathbf{h}_i, \mathbf{V}\mathbf{y}_{i-1}, \bar{\mathbf{x}}\right)$ for $\text{RNN}^m$. Note that the key difference between $\text{RNN}^b$ and $\text{RNN}^m$ is whether target labels are binary targets $y_i$ or 1-of-$L$ targets $\mathbf{y}_i$. Under the assumption that the hidden states $\mathbf{h}_i$ preserve the information on all previous labels $\boldsymbol{y}_{<i}$, learning $\text{RNN}^b$ and $\text{RNN}^m$ can be interpreted as learning classifiers in a chain. Whereas in PCCs an independent classifier is responsible for predicting each label, both proposed types of RNNs maintain a single set of parameters to predict all labels.

The input representations $\bar{\mathbf{x}}$ to both $\text{RNN}^b$ and $\text{RNN}^m$ are kept fixed after the preprocessing of inputs $\boldsymbol{x}$ is completed. Recently, an *encoder-decoder* (EncDec) framework, also known as *sequence-to-sequence* (Seq2Seq) learning [2, 28], has drawn attention to modeling both input and output sequences, and has been applied successfully to various applications in natural language processing and computer vision [5, 14]. EncDec is composed of two RNNs: an encoder network captures the information in the entire input sequence, which is then passed to a decoder network which decodes this information into a sequence of labels (Fig. 1d). In contrast to $\text{RNN}^b$ and $\text{RNN}^m$, which only use fixed input representations $\bar{\mathbf{x}}$, EncDec makes use of context-sensitive input vectors from $\boldsymbol{x}$. We describe how EncDec computes Eq. (3) in the following.

**Encoder.** An encoder takes $\boldsymbol{x}$ and produces a *sequence* of $D$-dimensional vectors $\mathbf{x} = \{\mathbf{x}_1, \mathbf{x}_2, \cdots, \mathbf{x}_E\}$ where $E$ is the number of encoded vectors for a single instance. In this work, we consider *documents* as input data. For encoding documents, we use words as atomic units. Consider a document as a sequence of $E$ words such that $\boldsymbol{x} = \{w_1, w_2, \cdots, w_E\}$ and a vocabulary of $\mathcal{V}$ words. Each word $w_j \in \mathcal{V}$ has its own $K$-dimensional vector representation $\mathbf{u}_j$. The set of these vectors constitutes a matrix of word embeddings defined as $\mathbf{U} \in \mathbb{R}^{K \times |\mathcal{V}|}$. Given this word embedding matrix $\mathbf{U}$, words in a document are converted to a sequence of $K$-dimensional vectors $\boldsymbol{u} = \{\mathbf{u}_1, \mathbf{u}_2, \cdots, \mathbf{u}_E\}$, which is then fed into the RNN to learn the sequential structures in a document

$$\mathbf{x}_j = \text{GRU}(\mathbf{x}_{j-1}, \mathbf{u}_j) \tag{4}$$

where $\mathbf{x}_0$ is the zero vector.

**Decoder.** After the encoder computes $\mathbf{x}_i$ for all elements in $\boldsymbol{x}$, we set the initial hidden state of the decoder $\mathbf{h}_0 = f_{\text{init}}(\mathbf{x}_E)$, and then compute hidden states $\mathbf{h}_i = \text{GRU}\left(\mathbf{h}_{i-1}, \mathbf{V}\mathbf{y}_{i-1}, \mathbf{c}_i\right)$ where $\mathbf{c}_i = \sum_j \alpha_{ij} \mathbf{x}_j$ is the context vector which is the sum of the encoded input vectors weighted by attention scores $\alpha_{ij} = f_{att}\left(\mathbf{h}_{i-1}, \mathbf{x}_j\right), \alpha_{ij} \in \mathbb{R}$. Then, as shown in [1], the conditional probability $P_\theta(\mathbf{y}_i | \boldsymbol{y}_{<i}, \boldsymbol{x})$ for predicting a label $\mathbf{y}_i$ can be estimated by a function of the hidden state $\mathbf{h}_i$, the previous label $\mathbf{y}_{i-1}$ and the context vector $\mathbf{c}_i$:

$$P_\theta(\mathbf{y}_i | \boldsymbol{y}_{<i}, \boldsymbol{x}) = f(\mathbf{h}_i, \mathbf{V}\mathbf{y}_{i-1}, \mathbf{c}_i). \tag{5}$$

Indeed, EncDec is potentially more powerful than $\text{RNN}^b$ and $\text{RNN}^m$ because each prediction is determined based on the dynamic context of the input $\boldsymbol{x}$ unlike the fixed input representation $\bar{\mathbf{x}}$ used

Table 1: Comparison of the three RNN architectures for MLC.

|  | $\text{RNN}^b$ | $\text{RNN}^m$ | EncDec |
|---|---|---|---|
| hidden states | GRU $\left(\mathbf{h}_{i-1}, \mathbf{V}_{y_{i-1}}, \bar{\mathbf{x}}\right)$ | GRU $\left(\mathbf{h}_{i-1}, \mathbf{V}\mathbf{y}_{i-1}, \bar{\mathbf{x}}\right)$ | GRU $\left(\mathbf{h}_{i-1}, \mathbf{V}\mathbf{y}_{i-1}, \mathbf{c}_i\right)$ |
| prob. of output labels | $f\left(\mathbf{h}_i, \mathbf{V}_{y_{i-1}}, \bar{\mathbf{x}}\right)$ | $f\left(\mathbf{h}_i, \mathbf{V}\mathbf{y}_{i-1}, \bar{\mathbf{x}}\right)$ | $f\left(\mathbf{h}_i, \mathbf{V}\mathbf{y}_{i-1}, \mathbf{c}_i\right)$ |

in PCC, $\text{RNN}^b$ and $\text{RNN}^m$ (cf. Figs. 1a to 1d). The differences in computing hidden states and conditional probabilities among the three RNNs are summarized in Table 1.

Unlike in the training phase, where we know the size of positive label set $T$, this information is not available during prediction. Whereas this is typically solved using a meta learner that predicts a threshold in the ranking of labels, EncDec follows a similar approach as [7] and directly predicts a virtual label that indicates the end of the sequence.

# 4 Experimental Setup

In order to see whether solving MLC problems using RNNs can be a good alternative to *classifier chain* (CC)-based approaches, we will compare traditional multi-label learning algorithms such as BR and PCCs with the RNN architectures (Fig. 1) on multi-label text classification datasets. For a fair comparison, we will use the same fixed label permutation strategies in all compared approaches if necessary. As it has already been demonstrated in the literature that label permutations may affect the performance of classifier chain approaches [23, 13], we will evaluate a few different strategies.

## 4.1 Baselines and Training Details

We use feed-forward NNs as a base learner of BR, LP and PCC. For PCC, beam search with beam size of 5 is used at inference time [13]. As another NN baseline, we also consider a feed-forward NN with binary cross entropy per label [21]. We compare RNNs to FastXML [22], one of state-of-the-arts in extreme MLC.[1] All NN based approaches are trained by using *Adam* [12] and dropout [26]. The dimensionality of hidden states of all the NN baselines as well as the RNNs is set to 1024. The size of label embedding vectors is set to 256. We used the NVIDIA Titan X to train NN models including RNNs and base learners. For FastXML, a machine with 64 cores and 1024GB memory was used.

## 4.2 Datasets and Preprocessing

We use three multi-label text classification datasets for which we had access to the full text as it is required for our approach EncDec, namely Reuters-21578,[2] RCV1-v2 [15] and BioASQ,[3] each of which has different properties. Summary statistics of the datasets are given in Table 2. For preparing the train and the test set of Reuters-21578 and RCV1-v2, we follow [21]. We split instances in BioASQ by year 2014, so that all documents published in 2014 and 2015 belong to the test set. For tuning hyperparameters, we set aside 10% of the training instances as the validation set for both Reuters-21578 and RCV1-v2, but chose randomly 50 000 documents for BioASQ.

The RCV1-v2 and BioASQ datasets provide label relationships as a graph. Specifically, labels in RCV1-v2 are structured in a tree. The label structure in BioASQ is a directed graph and contains cycles. We removed all edges pointing to nodes which have been already visited while traversing the graph using DFS, which results in a DAG of labels.

**Document Representations.** For all datasets, we replaced numbers with a special token and then build a word vocabulary for each data set. The sizes of the vocabularies for Reuters-21578, RCV1-v2 and BioASQ are 22 747, 50 000 and 30 000, respectively. *Out-of-vocabulary* (OOV) words were also replaced with a special token and we truncated the documents after 300 words.[4]

Table 2: Summary of datasets. # training documents ($N_{tr}$), # test documents ($N_{ts}$), # labels ($L$), label cardinality ($C$), # label combinations ($LC$), type of label structure (HS).

| DATASET | $N_{tr}$ | $N_{ts}$ | $L$ | $C$ | $LC$ | HS |
|---|---|---|---|---|---|---|
| Reuters-21578 | 7770 | 3019 | 90 | 1.24 | 468 | - |
| RCV1-v2 | 781 261 | 23 149 | 103 | 3.21 | 14 921 | Tree |
| BioASQ | 11 431 049 | 274 675 | 26 970 | 12.60 | 11 673 800 | DAG |

We trained *word2vec* [20] on an English Wikipedia dump to get 512-dimensional word embeddings **u**. Given the word embeddings, we created the fixed input representations $\bar{\mathbf{x}}$ to be used for all of the baselines in the following way: Each word in the document except for numbers and OOV words is converted into its corresponding embedding vector, and these word vectors are then averaged, resulting in a document vector $\bar{\mathbf{x}}$. For EncDec, which learns hidden states of word sequences using an encoder RNN, all words are converted to vectors using the pre-trained word embeddings and we feed these vectors as inputs to the encoder. In this case, unlike during the preparation of $\bar{\mathbf{x}}$, we do not ignore OOV words and numbers. Instead, we initialize the vectors for those tokens randomly. For a fair comparison, we do not update word embeddings of the encoder in EncDec.

## 4.3 Evaluation Measures

MLC algorithms can be evaluated with multiple measures which capture different aspects of the problem. We evaluate all methods in terms of both example-based and label-based measures.

*Example-based measures* are defined by comparing the target vector $\boldsymbol{y} = \{y_1, y_2, \cdots, y_L\}$ to the prediction vector $\hat{\boldsymbol{y}} = \{\hat{y}_1, \hat{y}_2, \cdots, \hat{y}_L\}$. *Subset accuracy* (ACC) is very strict regarding incorrect predictions in that it does not allow any deviation in the predicted label sets: ACC $(\boldsymbol{y}, \hat{\boldsymbol{y}}) = \mathbb{I}[\boldsymbol{y} = \hat{\boldsymbol{y}}]$. *Hamming accuracy* (HA) computes how many labels are correctly predicted in $\hat{\boldsymbol{y}}$: HA $(\boldsymbol{y}, \hat{\boldsymbol{y}}) = \frac{1}{L}\sum_{j=1}^{L}\mathbb{I}[y_j = \hat{y}_j]$. ACC and HA are used for datasets with moderate $L$. If $C$ as well as $L$ is higher, entirely correct predictions become increasingly unlikely, and therefore ACC often approaches $0$. In this case, the *example-based $F_1$-measure* (eb$F_1$) defined by Eq. (6) can be considered as a good compromise.

*Label-based measures* are based on treating each label $y_j$ as a separate two-class prediction problem, and computing the number of *true positives* ($tp_j$), *false positives* ($fp_j$) and *false negatives* ($fn_j$) for this label. We consider two label-based measures, namely *micro-averaged $F_1$-measure* (mi$F_1$) and *macro-averaged $F_1$-measure* (ma$F_1$) which are defined by Eq. (7) and Eq. (8), respectively.

$$\mathrm{eb}F_1\left(\boldsymbol{y}, \hat{\boldsymbol{y}}\right) \qquad \mathrm{mi}F_1 \qquad\qquad \mathrm{ma}F_1$$

$$= \frac{2\sum_{j=1}^{L} y_j \hat{y}_j}{\sum_{j=1}^{L} y_j + \sum_{j=1}^{L} \hat{y}_j} \quad (6) \qquad = \frac{\sum_{j=1}^{L} 2tp_j}{\sum_{j=1}^{L} 2tp_j + fp_j + fn_j} \quad (7) \qquad = \frac{1}{L}\sum_{j=1}^{L} \frac{2tp_j}{2tp_j + fp_j + fn_j} \quad (8)$$

mi$F_1$ favors a system yielding good predictions on frequent labels, whereas higher ma$F_1$ scores are usually attributed to superior performance on rare labels.

## 5 Experimental Results

In the following, we show results of various versions of RNNs for MLC on three text datasets which span a wide variety of input and label set sizes. We also evaluate different label orderings, such as frequent-to-rare (*f2r*), and rare-to-frequent (*r2f*), as well as a topological sorting (when applicable).

### 5.1 Experiments on Reuters-21578

Figure 2 shows the *negative log-likelihood* (NLL) of Eq. (3) on the validation set during the course of training. Note that as RNN$^b$ attempts to predict binary targets, but RNN$^m$ and EncDec make predictions on multinomial targets, the results of RNN$^b$ are plotted separately, with a different scale of the y-axis (top half of the graph). Compared to RNN$^m$ and EncDec, RNN$^b$ converges very slowly. This can be attributed to the length of the label chain and sparse targets in the chain since RNN$^b$ is trained to make correct predictions over all 90 labels, most of them being zero. In other words, the length of target sequences of RNN$^b$ is 90 and fixed regardless of the content of training documents.

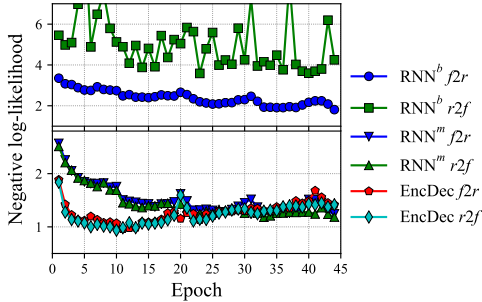

Figure 2: Negative log-likelihood of RNNs on the validation set of Reuters-21578.

Table 3: Performance comparison on Reuters-21578.

| | ACC | HA | $ebF_1$ | $miF_1$ | $maF_1$ |
|---|---|---|---|---|---|
| No label permutations | | | | | |
| BR(NN) | 0.7685 | 0.9957 | 0.8515 | 0.8348 | 0.4022 |
| LP(NN) | 0.7837 | 0.9941 | 0.8206 | 0.7730 | 0.3505 |
| NN | 0.7502 | 0.9952 | 0.8396 | 0.8183 | 0.3083 |
| Frequent labels first (*f2r*) | | | | | |
| PCC(NN) | 0.7844 | 0.9955 | 0.8585 | 0.8305 | 0.3989 |
| $RNN^b$ | 0.6757 | 0.9931 | 0.7180 | 0.7144 | 0.0897 |
| $RNN^m$ | 0.7744 | 0.9942 | 0.8396 | 0.7884 | 0.2722 |
| EncDec | **0.8281** | 0.9961 | 0.8917 | 0.8545 | **0.4567** |
| Rare labels first (*r2f*) | | | | | |
| PCC(NN) | 0.7864 | 0.9956 | 0.8598 | 0.8338 | 0.3937 |
| $RNN^b$ | 0.0931 | 0.9835 | 0.1083 | 0.1389 | 0.0102 |
| $RNN^m$ | 0.7744 | 0.9943 | 0.8409 | 0.7864 | 0.2699 |
| EncDec | 0.8261 | **0.9962** | **0.8944** | **0.8575** | 0.4365 |

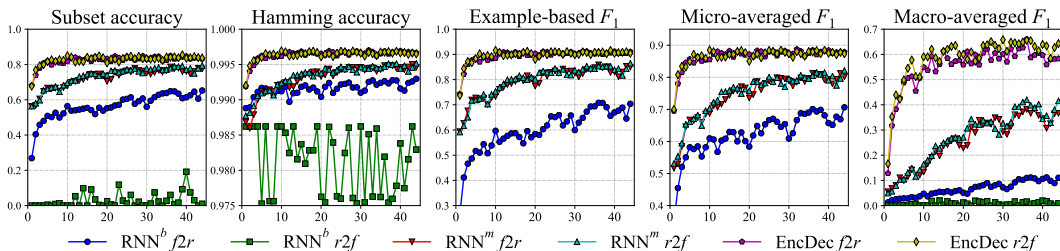

Figure 3: Performance of RNN models on the validation set of Reuters-21578 during training. Note that the x-axis denotes # epochs and we use different scales on the y-axis for each measure.

In particular, $RNN^b$ has trouble with the *r2f* label ordering, where training is unstable. The reason is presumably that the predictions for later labels depend on sequences that are mostly zero when rare labels occur at the beginning. Hence, the model sees only few examples of non-zero targets in a single epoch. On the other hand, both $RNN^m$ and EncDec converge relatively faster than $RNN^b$ and do obviously not suffer from the *r2f* ordering. Moreover, there is not much difference between both strategies since the length of the sequences is often 1 for Reuters-21578 and hence often the same.

Figure 3 shows the performance of RNNs in terms of all evaluation measures on the validation set. EncDec performs best for all the measures, followed by $RNN^m$. There is no clear difference between the same type of models trained on different label permutations, except for $RNN^b$ in terms of NLL (cf. Fig. 2). Note that although it takes more time to update the parameters of EncDec than those of $RNN^m$, EncDec ends up with better results. $RNN^b$ performs poorly especially in terms of $maF_1$ regardless of the label permutations, suggesting that $RNN^b$ would need more parameter updates for predicting rare labels. Notably, the advantage of EncDec is most pronounced for this specific task.

Detailed results of all methods on the test set are shown in Table 3. Clearly, EncDec perform best across all measures. LP works better than BR and NN in terms of ACC as intended, but performs behind them in terms of other measures. The reason is that LP, by construction, is able to more accurately hit the exact label set, but, on the other hand, produces more false positives and false negatives in our experiments in comparison to BR and NN when missing the correct label combination. As shown in the table, $RNN^m$ performs better than its counterpart, i.e., $RNN^b$, in terms of ACC, but has clear weaknesses in predicting rare labels (cf. especially $maF_1$). For PCC, our two permutations of the labels do not affect much ACC due to the low label cardinality.

## 5.2 Experiments on RCV1-v2

In comparison to Reuters-21578, RCV1-v2 consists of a considerably larger number of documents. Though the the number of unique labels ($L$) is similar (103 vs. 90) in both datasets, RCV1-v2 has a higher $C$ and $LC$ is greatly increased from 468 to 14 921. Moreover, this dataset has the interesting property that all labels from the root to a relevant leaf label in the label tree are also associated to the document. In this case, we can also test a topological ordering of labels, as described in Section 3.1.

Table 4: Performance comparison on RCV1-v2.

| | ACC | HA | $\mathrm{eb}F_1$ | $\mathrm{mi}F_1$ | $\mathrm{ma}F_1$ |
|---|---|---|---|---|---|
| No label permutations | | | | | |
| BR(NN) | 0.5554 | 0.9904 | 0.8376 | 0.8349 | 0.6376 |
| LP(NN) | 0.5149 | 0.9767 | 0.6696 | 0.6162 | 0.4154 |
| NN | 0.5837 | 0.9907 | 0.8441 | 0.8402 | 0.6573 |
| FastXML | 0.5953 | 0.9910 | 0.8409 | 0.8470 | 0.5918 |
| Frequent labels first (f2r) | | | | | |
| PCC(NN) | 0.6211 | 0.9904 | 0.8461 | 0.8324 | 0.6404 |
| $\mathrm{RNN}^m$ | 0.6218 | 0.9903 | 0.8578 | 0.8487 | 0.6798 |
| EncDec | **0.6798** | **0.9925** | 0.8895 | **0.8838** | 0.7381 |
| Rare labels first (r2f) | | | | | |
| PCC(NN) | 0.6300 | 0.9906 | 0.8493 | 0.8395 | 0.6376 |
| $\mathrm{RNN}^m$ | 0.6216 | 0.9903 | 0.8556 | 0.8525 | 0.6583 |
| EncDec | 0.6767 | **0.9925** | 0.8884 | 0.8817 | **0.7413** |
| topological sorting | | | | | |
| PCC(NN) | 0.6257 | 0.9904 | 0.8463 | 0.8364 | 0.6486 |
| $\mathrm{RNN}^m$ | 0.6072 | 0.9898 | 0.8525 | 0.8437 | 0.6578 |
| EncDec | 0.6761 | 0.9924 | 0.8888 | 0.8808 | 0.7220 |
| reverse topological sorting | | | | | |
| PCC(NN) | 0.6267 | 0.9902 | 0.8444 | 0.8346 | 0.6497 |
| $\mathrm{RNN}^m$ | 0.6232 | 0.9904 | 0.8561 | 0.8496 | 0.6535 |
| EncDec | 0.6781 | **0.9925** | **0.8899** | 0.8797 | 0.7258 |

Table 5: Performance comparison on BioASQ.

| | ACC | HA | $\mathrm{eb}F_1$ | $\mathrm{mi}F_1$ | $\mathrm{ma}F_1$ |
|---|---|---|---|---|---|
| No label permutations | | | | | |
| FastXML | 0.0001 | 0.9996 | 0.3585 | 0.3890 | 0.0570 |
| Frequent label first (f2r) | | | | | |
| $\mathrm{RNN}^m$ | 0.0001 | 0.9993 | 0.3917 | 0.4088 | 0.1435 |
| EncDec | 0.0004 | 0.9995 | 0.5294 | 0.5634 | 0.3211 |
| Rare labels first (r2f) | | | | | |
| $\mathrm{RNN}^m$ | 0.0001 | 0.9995 | 0.4188 | 0.4534 | 0.1801 |
| EncDec | 0.0006 | **0.9996** | 0.5531 | 0.5943 | 0.3363 |
| topological sorting | | | | | |
| $\mathrm{RNN}^m$ | 0.0001 | 0.9994 | 0.4087 | 0.4402 | 0.1555 |
| EncDec | 0.0006 | 0.9953 | 0.5311 | 0.5919 | **0.3459** |
| reverse topological sorting | | | | | |
| $\mathrm{RNN}^m$ | 0.0001 | 0.9994 | 0.4210 | 0.4508 | 0.1646 |
| EncDec | **0.0007** | **0.9996** | **0.5585** | **0.5961** | 0.3427 |

As $\mathrm{RNN}^b$ takes long to train and did not show good results on the small dataset, we have no longer considered it in these experiments. We instead include FastXML as a baseline.

Table 4 shows the performance of the methods with different label permutations. These results demonstrate again the superiority of PCC and $\mathrm{RNN}^m$ as well as EncDec against BR and NN in maximizing ACC. Another interesting observation is that LP performs much worse than other methods even in terms of ACC due to the data scarcity problem caused by higher $LC$. $\mathrm{RNN}^m$ and EncDec, which also predict label subsets but in a sequential manner, do not suffer from the larger number of distinct label combinations. Similar to the previous experiment, we found no meaningful differences between the $\mathrm{RNN}^m$ and EncDec models trained on different label permutations on RCV1-v2. FastXML also performs well except for $\mathrm{ma}F_1$ which tells us that it focuses more on frequent labels than rare labels. As noted, this is because FastXML is designed to maximize top-$k$ ranking measures such as prec@$k$ for which the performance on frequent labels is important.

## 5.3 Experiments on BioASQ

Compared to Reuters-21578 and RCV1-v2, BioASQ has an extremely large number of instances and labels, where $LC$ is almost close to $N_{tr} + N_{ts}$. In other words, nearly all distinct label combinations appear only once in the dataset and some label subsets can only be found in the test set. Table 5 shows the performance of FastXML, $\mathrm{RNN}^m$ and EncDec on the test set of BioASQ. EncDec clearly outperforms $\mathrm{RNN}^m$ by a large margin. Making predictions over several thousand labels is a particularly difficult task because MLC methods not only learn label dependencies, but also understand the context information in documents allowing us to find word-label dependencies and to improve the generalization performance.

We can observe a consistent benefit from using the reverse label ordering on both approaches. Note that EncDec does show reliable performance on two relatively small benchmarks regardless of the choice of the label permutations. Also, EncDec with reverse topological sorting of labels achieves the best performance, except for $\mathrm{ma}F_1$. Note that we observed similar effects with $\mathrm{RNN}^m$ in our preliminary experiments on RCV1-v2, but the impact of label permutations disappeared once we tuned $\mathrm{RNN}^m$ with dropout. This indicates that label ordering does not affect much the final performance of models if they are trained well enough with proper regularization techniques.

To understand the effectiveness of each model with respect to the size of the positive label set, we split the test set into five almost equally-sized partitions based on the number of target labels in the documents and evaluated the models separately for each of the partition, as shown in Fig. 4. The first partition (P1) contains test documents associated with 1 to 9 labels. Similarly, other partitions, P2, P3, P4 and P5, have documents with cardinalities of $10 \sim 12$, $13 \sim 15$, $16 \sim 18$ and more than 19, respectively. As expected, the performance of all models in terms of ACC and HA decreases as the

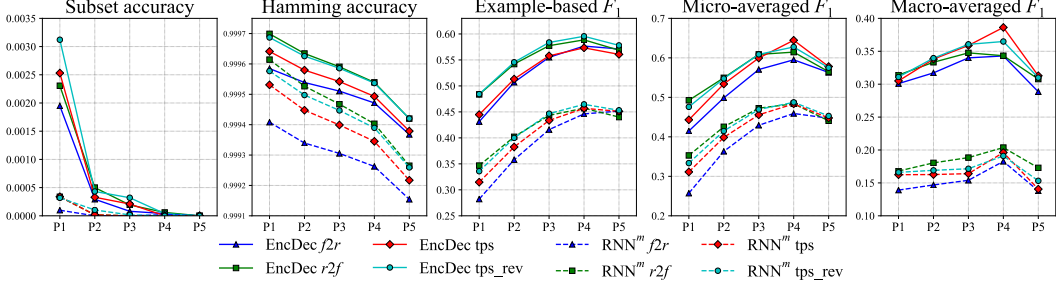

Figure 4: Comparison of RNN$^m$ and EncDec wrt. the number of positive labels $T$ of test documents. The test set is divided into 5 partitions according to $T$. The x-axis denotes partition indices. *tps* and *tps_rev* stand for the label permutation ordered by topological sorting and its reverse.

number of positive labels increases. The other measures increase since the classifiers have potentially more possibilities to match positive labels. We can further confirm the observations from Table 5 w.r.t. to different labelset sizes.

The margin of FastXML to RNN$^m$ and EncDec is further increased. Moreover, its poor performance on rare labels confirms again the focus of FastXML on frequent labels. Regarding computational complexity, we could observe an opposed relation between the used resources: whereas we ran EncDec on a single GPU with 12G of memory for 5 days, FastXML only took 4 hours to complete (on 64 CPU cores), but, on the other hand, required a machine with 1024G of memory.

## 6  Conclusion

We have presented an alternative formulation of learning the joint probability of labels given an instance, which exploits the generally low label cardinality in multi-label classification problems. Instead of having to iterate over each of the labels as in the traditional classifier chains approach, the new formulation allows us to directly focus only on the positive labels. We provided an extension of the formal framework of probabilistic classifier chains, contributing to the understanding of the theoretical background of multi-label classification. Our approach based on recurrent neural networks, especially encoder-decoders, proved to be effective, highly scalable, and robust towards different label orderings on both small and large scale multi-label text classification benchmarks. However, some aspects of the presented work deserve further consideration.

When considering MLC problems with extremely large numbers of labels, a problem often referred to as *extreme MLC* (XMLC), $F_1$-measure maximization is often preferred to subset accuracy maximization because it is less susceptible to the very large number of label combinations and imbalanced label distributions. One can exploit *General F-Measure Maximizer* (GFM) [30] to maximize the *example-based $F_1$-measure* by drawing samples from $P(\boldsymbol{y}|\boldsymbol{x})$ at inference time. Although it is easy to draw samples from $P(\boldsymbol{y}|\boldsymbol{x})$ approximated by RNNs, and the calculation of the necessary quantities for GFM is straightforward, the use of GFM would be limited to MLC problems with a moderate number of labels because of its quadratic computational complexity $\mathcal{O}(L^2)$.

We used a fixed threshold $0.5$ for all labels when making predictions by BR, NN and FastXML. In fact, such a fixed thresholding technique performs poorly on large label spaces. Jasinska et al. [10] exhibit an efficient *macro-averaged $F_1$-measure* (ma$F_1$) maximization approach by tuning the threshold for each label relying on the sparseness of $\boldsymbol{y}$. We believe that FastXML can be further improved by the ma$F_1$ maximization approach on BioASQ. However, we would like to remark that the RNNs, especially EncDec, perform well without any $F_1$-measure maximization at inference time. Nevertheless, ma$F_1$ maximization for RNNs might be interesting for future work.

In light of the experimental results in Table 5, learning from raw inputs instead of using fixed input representations plays a crucial role for achieving good performance in our XMLC experiments. As the training costs of the encoder-decoder architecture used in this work depend heavily on the input sequence lengths and the number of unique labels, it is inevitable to consider more efficient neural architectures [8, 11], which we also plan to do in future work.

**Acknowledgments**

The authors would like to thank anonymous reviewers for their thorough feedback. Computations for this research were conducted on the Lichtenberg high performance computer of the Technische Universität Darmstadt. The Titan X used for this research was donated by the NVIDIA Corporation. This work has been supported by the German Institute for Educational Research (DIPF) under the Knowledge Discovery in Scientific Literature (KDSL) program, and the German Research Foundation as part of the Research Training Group Adaptive Preparation of Information from Heterogeneous Sources (AIPHES) under grant No. GRK 1994/1.

## Footnotes

[1]Note that as FastXML optimizes top-$k$ ranking of labels unlike our approaches and assigns a confidence score for each label. We set a threshold of $0.5$ to convert rankings of labels into bipartition predictions.

[2]http://www.daviddlewis.com/resources/testcollections/reuters21578/

[3]http://bioasq.org

[4]By the truncation, one may worry about the possibility of missing information related to some specific labels. As the average length of documents in the datasets is below 300, the effect would be negligible.

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
