[Reviews · NeurIPS 2017]

Reviewer 1



I have been reviewing this paper for another conference, so in my review I mostly repeat my comments sent earlier. Already at that time I was for accepting the paper. It is worth to underline that the paper has been further improved by the authors since then. The paper considers a problem of solving multi-label classification (MLC) with recurrent neural networks (RNNs). MLC is converted into a sequential prediction problem: the model predicts a sequence of relevant labels. The prediction is made in a sequential manner taking into account all previous labels. This approach is similar to classifier chains, with two key differences: firstly, only single model is used, and its parameters are shared during prediction; secondly, RNN predicts "positive" labels only. In this way, the algorithm can be much more efficient than the standard classifier chains. The authors discuss also the problem of the right ordering of labels to be used during training and testing. The paper is definitely very investing. The idea of reducing MLC to sequential prediction problem is a natural extension of chaining commonly used for solving MLC under 0/1 loss. I suppose, however, that some similar ideas have been already considered. The authors could, for example, discuss a link between their approach and the learn-to-search paradigm used, for example, in "HC-Search for Multi-Label Prediction: An Empirical Study". I suppose that the algorithm presented in that paper could also be used for predicting sequences of "positive" labels. Minor comments: - The authors write that "Our first observation is that PCCs are no longer competitive to the sequence prediction approaches w.r.t. finding the mode of the joint distribution (ACC and maF1)". It seems however that PCC performs better than RNN under the ACC measure. - FastXML can be efficiently tuned for macro-F (see, "Extreme F-measure Maximization using Sparse Probability Estimates"). It is not clear, however, how much this tuning will improve the result on BioASQ. After rebuttal: I thank the authors for their response.

Reviewer 2



This paper presents a novel approach to learning the joint probability of labels in multi-label classification problems. The authors introduce a formulation where the joint probability is computed over a sequence of positive labels. The novel approach is analyzed empirically on three benchmark datasets. As claimed by the authors, the novel approach has a number of advantages compared to existing methods such as PCCs. One important advantage is that the length of the chain decreases by considering positive labels only, resulting in a more compact model that suffers less from error propagation. In addition, the use of seq-to-seq neural nets instead of more traditional base learners in PCC can also result in performance gains in areas where deep learning is typically successful (images, text, speech, etc.). In general I find the method proposed in this paper interesting. However, I do see a few possibilities for further improving the paper: - In the beginning of the paper the authors indicate that they are interested in minimizing the subset zero-one loss. This might be a natural choice, given that a joint probability distribution over labels is constructed. However, in the experiments, the authors start to focus on other loss functions for which the proposed framework is not suited. It is clear that a BR-style method should be preferred for Hamming loss, whereas more complicated approaches are needed to optimize the different variants of the F-measure. The analyzed methods, neither the PCC baseline, nor the method of the authors, optimize the F-measure. More complicated algorithms are needed for that, see e.g. Waegeman et al. On the Bayes optimality of F-measure maximizers, JMLR 2014. - In light of this discussion, it is also a bit strange to optimize subset zero-one loss in MLC applications with hundreds of labels. Subset zero-one loss is for sure not the measure of interest in extreme MLC applications.